# Data-driven regularization lowers the size barrier of cryo-EM structure determination

Dari Kimanius[1,2] ✉, Kiarash Jamali[1], Max E. Wilkinson[3,4,5], Sofia Lövestam[1], Vaithish Velazhahan[1,6], Takanori Nakane[7] & Sjors H. W. Scheres [1] ✉

Macromolecular structure determination by electron cryo-microscopy (cryo-EM) is limited by the alignment of noisy images of individual particles. Because smaller particles have weaker signals, alignment errors impose size limitations on its applicability. Here, we explore how image alignment is improved by the application of deep learning to exploit prior knowledge about biological macromolecular structures that would otherwise be difficult to express mathematically. We train a denoising convolutional neural network on pairs of half-set reconstructions from the electron microscopy data bank (EMDB) and use this denoiser as an alternative to a commonly used smoothness prior. We demonstrate that this approach, which we call Blush regularization, yields better reconstructions than do existing algorithms, in particular for data with low signal-to-noise ratios. The reconstruction of a protein–nucleic acid complex with a molecular weight of 40 kDa, which was previously intractable, illustrates that denoising neural networks will expand the applicability of cryo-EM structure determination for a wide range of biological macromolecules.

Despite rapid progress in cryo-EM technology in the past decade[1], many biological macromolecules of interest are still too small to allow reliable structure determination. To limit the damage that electrons cause to biological structures of interest, cryo-EM images are taken using low doses of electron radiation, leading to high levels of experimental noise. The noise in the images impedes their alignment, resulting in an ill-posed optimization problem in which many reconstructions (which might be noisy or artifactual) are equally probable, given the data. The ill-posedness of the reconstruction imposes a minimum size barrier for cryo-EM structure determination, because smaller complexes yield images with lower signal-to-noise ratios. Although this barrier has been overcome in experiments involving the formation of complexes between small targets and other proteins[2], the formation of sufficiently rigid complexes is often difficult. Here we explore a computational method that lowers the size barrier for existing cryo-EM datasets.

Even for ill-posed reconstruction problems, the correct solution can still be identified through the incorporation of prior knowledge.

Most cryo-EM structures are calculated using explicit regularization of a likelihood function in Fourier space, which assumes cryo-EM reconstructions are smooth in real space[3–5]. Although we know much more about the structures of biological macromolecules beyond just the fact that their density varies smoothly, it has been difficult to incorporate richer sources of prior knowledge into the optimization process. Denoising convolutional neural networks can incorporate complex prior knowledge into an iterative optimization process[6]. By training a denoising network on simulated pairs of noisy and ground-truth images, we have previously provided proof of principle that prior knowledge about protein structures can be exploited to improve cryo-EM structure determination[7]. However, we also observed problems with overfitting and the hallucination of protein-like features in the resulting reconstructions. Moreover, because experimental cryo-EM structures often comprise regions of well-ordered proteins and nucleic acid domains alongside less structured regions, including, for example, membrane patches or flexible domains, it was not

[1]MRC Laboratory of Molecular Biology, Francis Crick Avenue, Cambridge, UK. [2]CZ Imaging Institute, Redwood City, CA, USA. [3]Broad Institute of MIT and Harvard, Cambridge, MA, USA. [4]McGovern Institute for Brain Research, Massachusetts Institute of Technology, Cambridge, MA, USA. [5]Howard Hughes Medical Institute, Cambridge, MA, USA. [6]School of Medicine, Stanford University, Stanford, CA, USA. [7]Institute for Protein Research, Osaka University, Suita-shi, Osaka, Japan. ✉e-mail: dari.kimanius@czii.org; scheres@mrc-lmb.cam.ac.uk

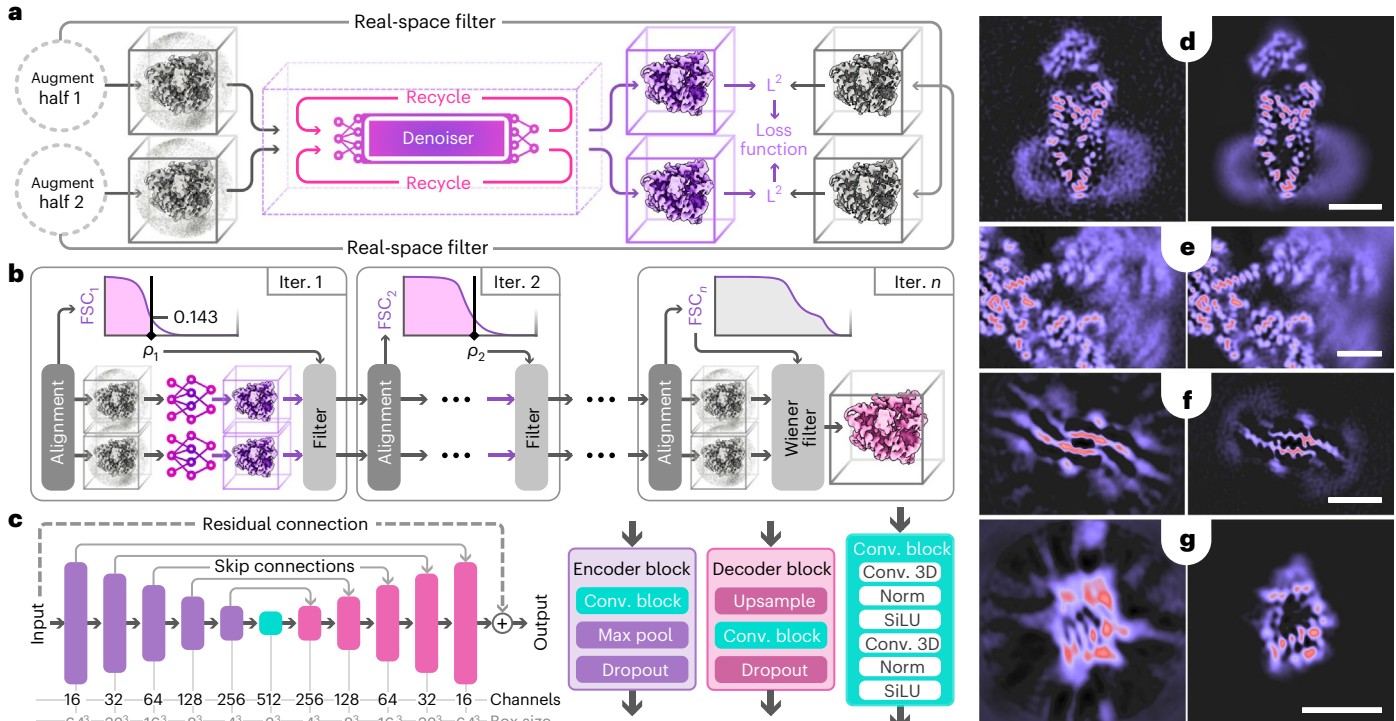

**Fig. 1 | Schematic illustration of Blush regularization and slices of example volumes. a**, Training procedure, showing two passes for both half-maps and recycling of the denoiser output (in pink), with calculation of a mean squared error (L2) loss. **b**, Iterative reconstruction with spectral trailing. Each half-map is reconstructed separately. At each iteration, the FSC is used to estimate a cut-off frequency ($\rho$), which is subsequently used to low-pass filter the denoiser output. The final output does not pass through the denoiser but is subjected to a Wiener filter, similar to baseline reconstruction. **c**, Denoiser U-net architecture,

consisting of five consecutive encoder blocks and a convolution block, followed by five consecutive decoder blocks. SiLU stands for sigmoid linear unit; Norm for batch normalization. **d,e**, Slices through maps before (left) and after (right) a single application of the denoiser to the final iteration of the reconstruction for PfCRT (**d**) and the spliceosome (**e**). **f,g**, Slices through maps of baseline reconstruction (left) and after Blush regularization (right) of the FIA (**f**) and the Aca2–RNA complex (**g**). Scale bars, 30 Å.

clear how ground-truth pairs for experimental cryo-EM data could be generated.

Here, we demonstrate how a pre-trained denoising convolutional neural network, trained and deployed in an application-specific manner inspired by the noise2noise approach[8] (Fig. 1 and Methods), can improve cryo-EM structure determination using experimental data. Through this approach, which we call Blush regularization, we improve reconstructions across a variety of existing cryo-EM datasets, including one for a protein–nucleic acid complex that was too small for analysis using existing methods.

## Results

### Blush regularization improves reconstruction without overfitting

We first tested Blush regularization on a cryo-EM dataset (EMPIAR-10330)[9] for the *Plasmodium falciparum* chloroquine resistance transporter (PfCRT)[10]. This dataset has been used as a standard to demonstrate the performance of several approaches in reducing overfitting during cryo-EM refinement[11,12]. Standard refinement using regularized likelihood optimization in RELION, which we refer to as the baseline, yields an overall resolution of 3.8 Å for this data set.

Application of Blush regularization (Fig. 2) yielded an overall resolution estimate of 3.4 Å. In the last iteration, spectral trailing, a heuristic method that prevents overfitting by limiting the spatial frequency at which information from the denoiser is used (Methods), was applied with a cut-off at 3.5 Å. Compared with the baseline reconstruction, local resolution improved for most regions of the map, with a corresponding increase in visible side-chain densities. The improvement in resolution, as measured by half-map Fourier shell correlation (FSC), was

confirmed by FSCs between both maps and the atomic model that was deposited for this dataset (Protein Data Bank (PDB): 6UKJ). Throughout this paper, FSCs between the map and atomic model were calculated using Servalcat[13]. We also assessed the relative quality of both maps by application of our automated model-building software ModelAngelo[14], which generated a model with 84% completeness in the baseline map and 97% completeness in the Blush map. Model completeness is defined as the percentage of residues that match the reference model with a Cα distance of 3 Å or less.

To assess the potential for overfitting by the denoiser, we also performed a phase-randomization test[15]. We applied Blush regularization without spectral trailing for refinement of the PfCRT dataset with phase randomization beyond 4-Å resolution. Although spectral trailing was not used, no overfitting was observed. Switching off spectral trailing led to a marginal improvement in the quality of reconstruction, as quantified by the FSC between the map and the atomic model (Fig. 2d). These results indicate that the denoiser can prevent overfitting for this dataset, even without spectral trailing. In general, we still recommend running Blush regularization with spectral trailing, because the benefits of switching it off are small and overfitting could be more prominent for other datasets. Consequently, in the following sections, we present results obtained only using spectral trailing.

### Blush expands the applicability of cryo-EM reconstruction

We subsequently assessed the broader applicability of Blush regularization by applying it to four types of structures and refinement methods.

First, we tested Blush regularization on a small membrane protein, Ste2, which is a dimeric G-protein-coupled receptor (GPCR)[16] (Fig. 3 and Extended Data Table 1). Full-length monomeric Ste2 has a molecular

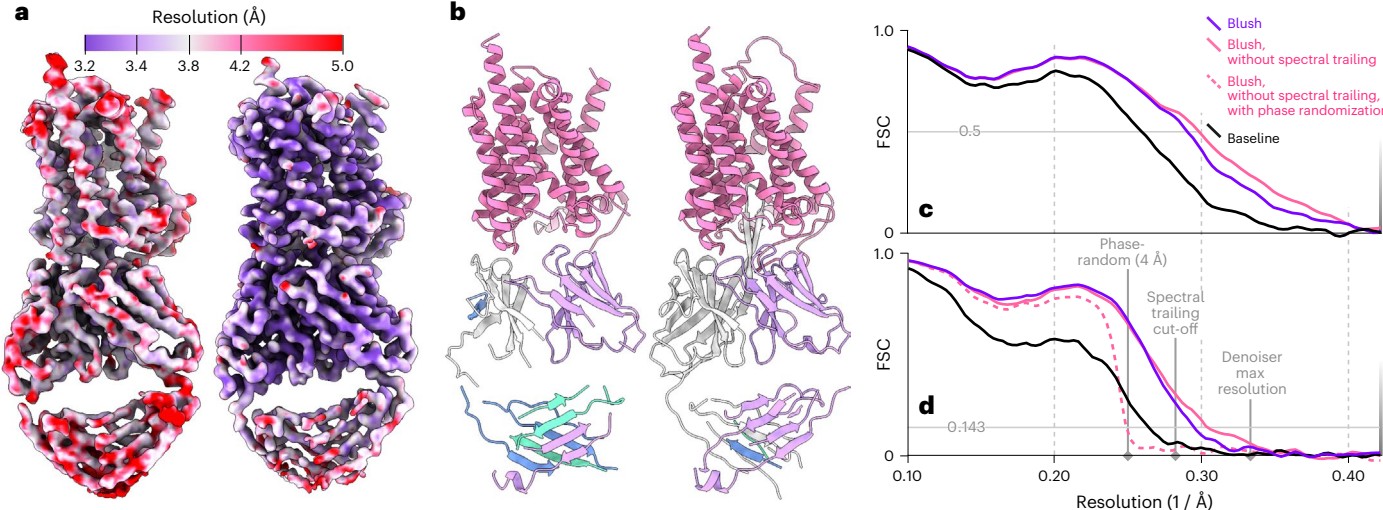

**Fig. 2 | Single-particle reconstruction of the PfCRT dataset. a**, Maps colored by local resolution, comparing the baseline reconstruction (left) and the reconstruction after Blush regularization (right). **b**, Automated atomic modeling by ModelAngelo for the baseline (left) and Blush (right) maps. Colored by chain. **c**, FSCs between the masked maps and deposited model (PDB: 6UKJ). **d**, Solvent-corrected half-map FSCs. Both plots show FSCs for Blush (purple), Blush without spectral trailing (pink) and baseline (black). The dashed pink line shows the solvent-corrected half-map FSC for Blush without spectral trailing when applied to data with phase randomization beyond 4-Å resolution.

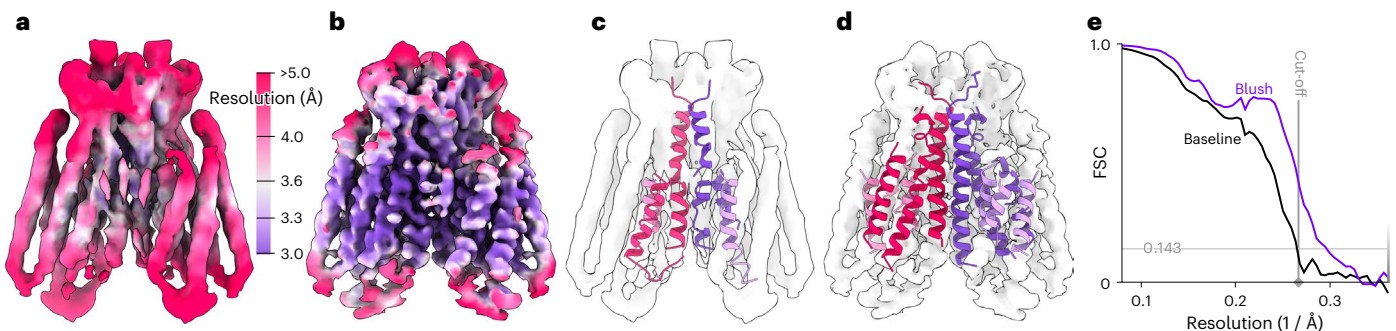

**Fig. 3 | Single-particle reconstruction of the Ste2 dataset. a,b**, Reconstructions of the Ste2 dataset, colored by local resolution, comparing the baseline reconstruction (**a**) and the reconstruction after Blush regularization (**b**). **c,d**, Automated atomic modeling by ModelAngelo, using the baseline (**c**) and Blush (**d**) maps. **e**, Solvent-corrected half-map FSCs.

weight of 47.85 kDa, which includes a long disordered carboxy-terminal tail that comprises 125 amino acids. The total mass of the ordered dimeric Ste2 that contributes to alignment is roughly 67 kDa, most of which lies embedded in a detergent micelle.

The dataset used was acquired from a similar complex to that in PDB entry 7QB9, reported in ref. 16, but with different biochemical conditions affecting the stability of the structure. Alignment of images of Ste2 is difficult because few protein features extend from the smooth detergent micelle. Baseline reconstruction yielded a map with an overall resolution of 3.8 Å, with limited densities for side chains. Application of Blush regularization led to a structure with an overall resolution of 3.4 Å. Spectral trailing ensured that no information from the denoiser was inserted beyond 3.7-Å resolution. Compared with the baseline reconstruction, the density of the transmembrane helices is improved. Loops at the top and bottom of the structure are still relatively poorly resolved, probably owing to molecular flexibility. In agreement with the visibility of improved side-chain densities and local resolution estimates, the completeness of models built by ModelAngelo in these maps improved from 19% to 43%.

Second, we evaluated the performance of Blush regularization in multi-body refinement[17], in which partial signal subtraction is used to align independently moving domains within a larger complex.

Reconstructions from subtracted images were included in the training set for the denoiser. Moreover, signal subtraction reduces the amount of signal in each image, placing stringent limitations on the minimal size of domains that can be aligned. We applied Blush regularization in multi-body refinement of a publicly available dataset (EMPIAR-10180) for the *Saccharomyces cerevisiae* pre-catalytic spliceosomal B complex[18] (Fig. 4). Using four bodies, one each for the core, the foot, the helicase and the SF3b regions, Blush regularization improved the quality of reconstructions of all domains compared with baseline multi-body refinement, as measured by local resolution, half-map FSCs and FSCs with the reference atomic model (PDB: 5NRL). The improvements in resolution were largest in the helicase and SF3b regions, which are the most flexible and thus the hardest to reconstruct. The improvements in resolution were reflected by automated model building in ModelAngelo, which increased model completeness of the entire complex from 32% to 48%. In particular, the model completeness for the SF3b region was improved from 3% to 29%.

Third, we assessed the performance of Blush regularization for a biological assembly that was different than the types of structures that the denoiser was trained on: the first intermediate amyloid (FIA) that forms during the in vitro assembly of recombinant tau (residues 297–391)[19]. This dataset is also publicly available (EMPIAR-11720). Unlike

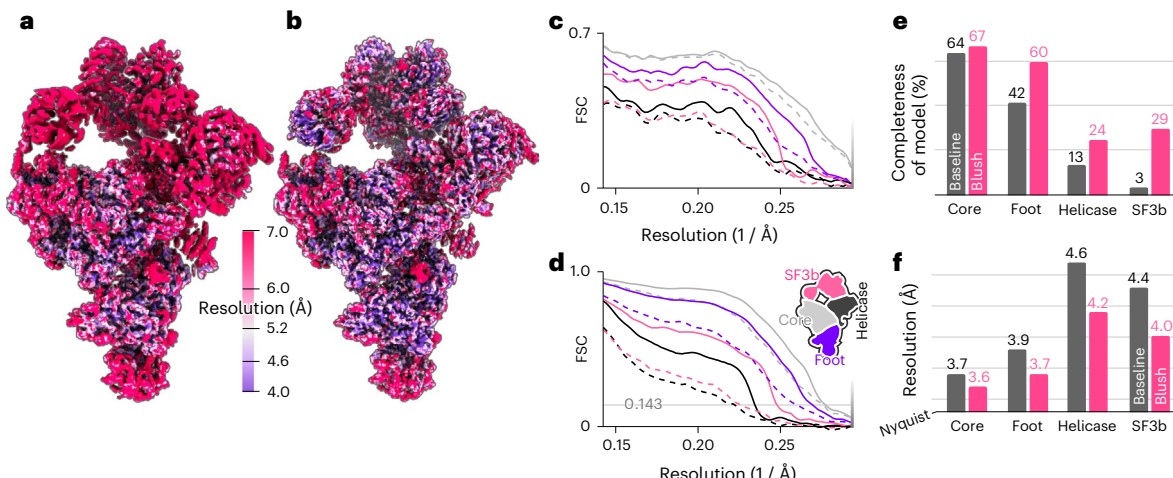

**Fig. 4 | Multi-body reconstruction of the spliceosome dataset. a,b,** Combined maps of the individual bodies, colored by local resolution, comparing the baseline reconstruction (**a**) and the reconstruction after Blush regularization (**b**). **c,** FSCs between the masked maps of each body and the corresponding region in the deposited model (PDB: 5NRL). **d,** Solvent-corrected half-map FSCs for the individual bodies. In **c** and **d**, dashed and solid lines correspond to baseline and Blush maps, respectively. FSCs are shown for each body: core (light gray), foot (dark gray), helicase (purple) and SF3b (pink). **e,** Completeness of atomic models built by ModelAngelo for each body, using baseline (gray) and Blush (pink) maps. **f,** Gold-standard half-map resolutions of each body for baseline (gray) and Blush (pink) maps.

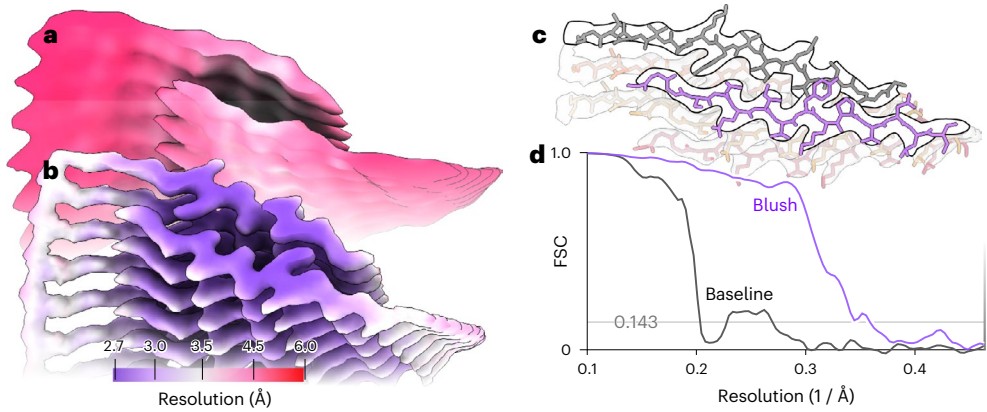

**Fig. 5 | Helical reconstruction of the FIA, colored by local resolution, for the baseline. a,b,** Maps of the baseline reconstruction (**a**) and after application of Blush regularization (**b**). **c,** Automated atomic modeling by ModelAngelo, comprising tau residues 302–316. **d,** Solvent-corrected half-map FSCs of the reconstructed maps.

any of the structures in the training set, the FIA has helical symmetry (Fig. 5). It is an amyloid filament, with parallel β-strands repeating every 4.7 Å in the direction of the helical axis. Besides deviating from the types of structures in the training set, the FIA is also one of the smallest amyloid structures solved to date, with only 15 ordered residues in each of two opposing β-sheets. Baseline helical refinement yielded a 5.0-Å-resolution map, in which the density for β-strands along the helical axis was not separated, and no atomic model could be built. Blush regularization improved the resolution to 2.8 Å, and ModelAngelo built all 15 ordered residues in the resulting map.

Fourth, we applied Blush to the small anti-CRISPR associated protein 2 (Aca2) bound to RNA, which has a total molecular weight of 40 kDa (Fig. 6 and Extended Data Table 1). Using different classification and refinement strategies in baseline RELION and CryoSPARC, we could not obtain a reliable reconstruction. Although an initial model generated using the standard VDAM algorithm in RELION[20] suffered from anisotropy, the first three-dimensional (3D) classification using Blush regularization resulted in one class with recognizable protein features. Similar 3D classifications without Blush regularization did not yield recognizable protein features. Refinement of the corresponding class yielded a better initial model for a second 3D classification, from which

a single class was selected for subsequent CTF refinement[21] and particle polishing[21]. A 3D classification was performed without alignment, followed by a final 3D refinement. Blush regularization was used for all 3D classifications with alignment and 3D refinements. The resolution of the final map was 2.5 Å, with ModelAngelo successfully building 97% of the protein sequence and 33 out of 42 nucleotides.

## Discussion

In a previous approach using noise2noise, implemented in the M software[22], a new neural network is trained for each dataset that it is applied to, using only half-maps from the same dataset. As such, the neural network in the M software can learn only features that are specific to the dataset at hand. By contrast, we pre-train a single neural network on a diverse set of high-resolution half-maps from the EMDB. Our pre-trained network improves cryo-EM reconstructions for a wide variety of macromolecular complexes, suggesting that it has learned useful features about cryo-EM structures in general. In addition, although our approach was inspired by noise2noise, it blends the unsupervised elements from noise2noise training with new application-specific elements, such as recycling and supervised masks in Fourier space and in real space. An interesting avenue for future research could be

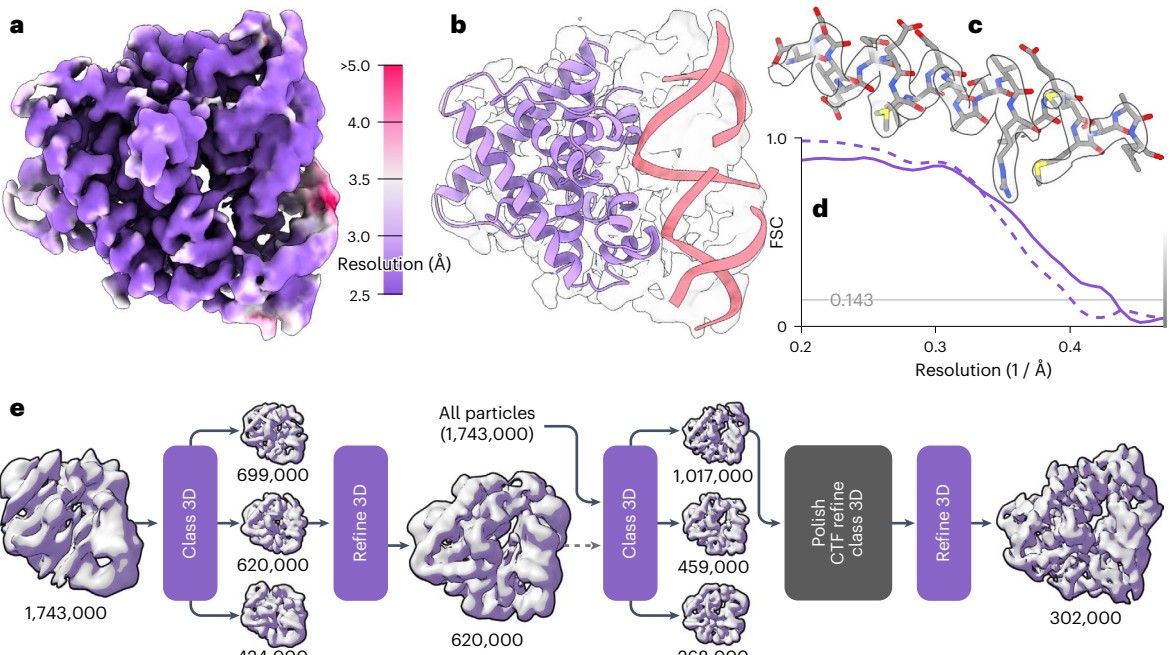

**Fig. 6 | Single-particle reconstruction of the Aca2–RNA complex with a molecular weight of 40 kDa. a**, Local resolution of the reconstruction with Blush regularization. **b**, Automated atomic model assignment with ModelAngelo. **c**, Detailed view of an α-helical segment in the reconstructed map and refined atomic model. **d**, The solid line shows the FSC to a reference atomic model, and the dashed line shows the half-map FSC. The solvent-corrected resolution is 2.5 Å, using a spectral trailing cut-off at 3.0 Å. **e**, Processing pipeline from initial model to final reconstruction. Numbers indicate the number of particles assigned to each map. Purple squares indicate reconstructions using Blush regularization.

a combination of the two approaches, in which the pre-trained Blush network is fine tuned using the half-maps of the dataset at hand, using techniques similar to those implemented in M.

We previously attempted to incorporate prior knowledge about protein structures by training a denoiser on pairs of noisy and ground-truth maps that were calculated from atomic models, and observed problems with overfitting and hallucinations[7]. Similar problems could explain why the application of the DeepEMhancer neural network[23] inside the iterative reconstruction algorithm of RELION had to be restricted to only a few iterations at the end of refinement[24]. The approach in this paper reduces the risk of hallucinations of protein-like features in reconstructions by using a neural network that is trained only on experimental cryo-EM half-maps, that is without the atomic models or the geometrical restraints that are used to describe them.

Instead of forcing the map to resemble densities derived from atomic models, our denoiser is trained to introduce more subtle modifications to cryo-EM maps, such as smoothing out density in solvent regions or in detergent micelles. The network also removes artifacts that are commonly encountered in difficult cryo-EM refinements, for example anisotropic densities that result from uneven angular distributions, or radially extending, streaky features that are often observed in overfitted maps (Figs 1f,g). Our findings illustrate that, although the effect of a single application of the denoiser is relatively small, its cumulative impact over several iterations enhances the performance of cryo-EM structure determination across a diverse range of test cases. As the ability of machine-learning methods to extract knowledge from large datasets improves, it could be tempting to leverage more structural information about biological macromolecules in the reconstruction process. However, doing so could ultimately diminish one of the most powerful ways of assessing whether a reconstruction is correct: the presence of expected features in the map. We thus anticipate that the cryo-EM community will continue to explore the question of how much prior knowledge should inform the reconstruction process, and how much should be kept aside for validation.

In the framework of Blush regularization, the denoiser replaces the filter operation that constrains the power of Fourier-space components in the baseline algorithm. As a result, the FSC between independently refined subsets is no longer used to define a 3D Wiener filter that is applied to the intermediate reconstructions. Instead, this FSC is used to determine a resolution cut-off ($\rho$), beyond which the Fourier components of the two denoised half-maps are set to zero. Because Fourier components near the resolution estimate of the final map will not have been affected by the denoiser, overestimation of resolution owing to the denoiser cannot happen directly.

Although spectral trailing represents the first attempt to prevent overestimation of resolution when using information-rich priors in cryo-EM reconstruction, it might not be the optimal solution. In fact, as exemplified by the PfCRT dataset (Fig. 2), spectral trailing can lead to underestimation of resolution. Future exploration of the damping effect of the network in Fourier space could lead to better approaches to safeguard against overestimation of resolution. Other research topics that might be worth exploring include the adaptation of the VDAM algorithm[20] in Relion to also use Blush regularization, which may improve initial model generation. In fact, provided that they allow modification of real-space maps, a wide range of cryo-EM methods could be improved by Blush regularization, ranging from standard refinement approaches in alternative software packages to approaches for dealing specifically with structural heterogeneity, for example[25–27].

In all our tests, the performance of Blush regularization surpassed or matched that of the baseline implementation in RELION. We observed the largest differences for cases in which the baseline approach tended to overfit the data. Consequently, Blush regularization will be most useful for refinements of datasets with low signal-to-noise ratios, such as those of small complexes or complexes embedded in thick ice layers, multi-body refinements involving relatively small bodies and refinements of maps exhibiting pronounced variations in local resolution. For example, Blush regularization allowed reconstruction of an amyloid with only 30 residues in its ordered core,

and of the Aca2–RNA complex with a molecular weight of 40 kDa. Although nucleic acids result in higher signal-to-noise ratios than do proteins, 40 kDa approaches predicted minimal sizes for a protein that is amenable to cryo-EM structure determination[28,29]. These results demonstrate that denoising convolutional neural networks expands the applicability of cryo-EM structure determination .

## Online content

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

## Methods

### Rationale

The noise2noise framework[8] facilitates the training of a denoising convolutional neural network in the absence of explicit access to ground-truth images. Instead, it relies on pairs of noisy images to extract information about their shared signal. Here, we present an application-specific approach that incorporates this aspect from the noise2noise framework. We trained a denoiser on a set of 422 pairs of noisy half-maps that we downloaded from the EMDB[30]. We selected only entries with reported resolutions higher than 4 Å for which both unfiltered half-maps were deposited. Maps with obvious artifacts, for example those associated with overfitting, and maps of a structure that was already present in the training set were eliminated during manual curation.

We tailored data augmentation and training of the denoiser to integrate with the iterative expectation-maximization algorithm for cryo-EM reconstruction. All pairs of half-maps, $x_i^{(k)} \in \mathbb{R}^N$, with $k \in \{0,1\}$, were re-scaled to a uniform voxel size of 1.5 Å, and augmented by generating new pairs $y_i^k, \bar{y}_i^k \in \mathbb{R}^N$:

$$y_i^{(k)} = H_{C,A}\left[x_i^{(1-k)} + e^{(1-k)}\right], \tag{1}$$

$$\bar{y}_i^{(k)} = H_{\bar{C},A}\left[x_i^{(k)} \odot M_i + h\left(x_i^{(k)}\right) \odot (1 - M_i)\right], \tag{2}$$

where $e \in \mathbb{R}^N$ is random colored noise, $M_i \in [0,1]^N$ is a smooth mask encapsulating the molecules of interest, $\odot$ represents voxel-wise multiplication and h(.) is a low-pass filter to 15 Å. $H_{C,A}[.]$ applies an anisotropic Gaussian filter with covariance matrix $C$, an affine transform $A$ that includes rotation and translation, a crop to a patch of $64^3$ voxels and a voxel-value standardization. Data augmentation was achieved through random assignments of $C, \bar{C}, A, e$ and $r$.

By using a range of resolution cut-offs for $C$ and $\bar{C}$, the denoiser explicitly learns to handle maps with varying resolutions. This is necessary for its application inside the iterative expectation-maximization algorithm, which typically starts at relatively low resolutions and gradually progresses to higher resolutions. Although using a lower resolution cut-off for $C$ than for $\bar{C}$ could have produced a network that enhances the resolution of the half-maps, similar to deblurring networks[31], we opted not to do so to minimize the risk of hallucinations in high-resolution features.

Using different degrees of anisotropy in $C$ and $\bar{C}$, the denoiser learns to deal with the artifacts that arise from non-uniform orientational distributions, and random orientations and affine transformations in $A$ lead to invariance with respect to rotations, translations and intensity scale. Although initial versions of our training protocol did not include masks, we observed that the resulting networks would learn to smoothen densities in disordered regions, such as the solvent or detergent micelles, which would improve image alignments. To amplify these effects, we then implemented the supervised masking approach with $M_i$ and h(.). By filling disordered regions with a 15-Å-resolution low-pass filtered version of the map, as opposed to a straightforward voxel-wise multiplication with the mask $M_i$, higher density values in regions with disordered molecules, such as detergent micelles, are maintained.

By re-scaling all maps to a common voxel size of 1.5 Å, and then cropping maps to patches of $64^3$ voxels, the network can be trained on and applied to maps of any size. To apply the denoiser to maps that are larger than one patch, overlapping patches can be denoised independently.

### Training the denoiser

Our denoiser ($f_\theta$) consists of a U-net with approximately 13 million trainable parameters ($\theta$) (Fig. 1). It is trained using residual learning[32] and with a dropout rate of 50% (ref. 33). Instance normalization[34] is used to handle small mini-batches ($\mathcal{B}$), with $b = 8$ samples from the training dataset, during training. We minimize the following loss:

$$\mathcal{L} = \frac{1}{2b}\sum_{i \in \mathcal{B}}\sum_{k \in \{0,1\}}\left\| \bar{y}_i^{(k)} - f_\theta\left(R_r\left[f_\theta, y_i^{(k)}\right]\right)\right\|^2, \tag{3}$$

where $R_r[f_\theta, y]$ returns the output of the denoiser $f_\theta$ after recursively calling it $r \in \{0, ..., 5\}$ times with $y_i^k$ as the initial input. This enables the denoiser to recognize and suppress artifacts brought about by its repeated usage, thereby limiting the amplification of artifacts in the reconstruction that are introduced by the denoiser during subsequent iterations of the expectation-maximization algorithm[7].

Training for 950,000 steps took six days using a single Nvidia A100 GPU.

### Iterative denoising with spectral trailing

We refer to the application of our pre-trained denoiser within the iterative expectation-maximization algorithm as Blush regularization. In our original work, with simulated data, we incorporated the denoiser into the $L_2$ regularization in the M-step, on the basis of the approximation that the prior function is 'close' to a Gaussian[7]. In this work, we do not make formal claims about the role of the denoiser within a Bayesian framework. Instead, our approach is motivated by empirical observations.

Although one effect of the denoiser is that it tends to dampen Fourier components at higher spatial frequencies, the amount by which it does so is not well defined. Therefore, we use a heuristic method, here referred to as spectral trailing, to prevent overfitting in 3D autorefinement and multi-body refinement. First, we calculate the FSC between two independently refined half-maps before the denoiser is applied, and determine the $\rho$ value at which the solvent-corrected FSC drops below 0.143. We then apply the denoiser to both half-maps and subsequently apply a low-pass filter at a spatial frequency that is two Fourier shells (each shell is one Fourier voxel wide) lower than $\rho$. If $\rho$ exceeds the Nyquist frequency of the denoiser, here set to 3 Å, the remaining Fourier shells at higher frequencies are populated with the reconstruction from the standard regularization in Fourier space. The resulting denoised, low-pass-filtered maps are then used as references for alignment in the next iteration. The denoiser is not applied to the output of the final refinement step.

Blush regularization has been implemented in the open-source software RELION-5, using a combination of C++ and PyTorch. It can be used for 3D classification, multi-body refinement and 3D autorefinement jobs, including those for particles with point-group or helical symmetry. For 3D classification for data that are separated into independent half-sets, the filtered map from the regularized likelihood approach is used as input for the denoiser. No additional low-pass filtering is applied. In this job type, the denoiser is also applied in the last iteration.

### Reporting summary

Further information on research design is available in the Nature Portfolio Reporting Summary linked to this article.

## Data availability

The full list of EMDB entries that were used to train the denoiser, along with the manually curated masks, can be downloaded from https://zenodo.org/records/10553452 (ref. 35). The Aca2–RNA dataset has been submitted to EMPIAR (EMPIAR-11918).

## Code availability

Blush regularization has been implemented in the open-source software RELION-5, which is distributed for free under the GPLv2 license and can be downloaded from https://github.com/3dem/relion. Additionally, code used in the training procedure of the Blush denoiser model is available at https://github.com/dkimanius/blush-training.

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

## Acknowledgements

We thank J. Schwab, K. Yamashita, C.-B. Schönlieb and O. Öktem for helpful discussions; J. Grimmett, T. Darling and I. Clayson for help with high-performance computing; the EM facility at the Medical Research Council Laboratory of Molecular Biology for support with cryo-EM; N. Birkholz and P. Fineran for input into the design and production of Aca2–RNA complexes, funded by Bioprotection Aotearoa, Centre of Research Excellence (Tertiary Education Commission, New Zealand); and E. Brignole and C. Borsa for the smooth running of the MIT. nano cryo-EM facility, established in part with financial support from the Arnold and Mabel Beckman Foundation. M.E.W. is grateful to F. Zhang for funding support. T.N. is a member of the JEOL YOKOGUSHI Research Alliance Laboratories. This work was supported by the Medical Research Council as part of the United Kingdom Research and Innovation (MC_UP_A025_1013 to S.H.W.S.); the European Union's Horizon 2020 research and innovation program (under grant agreement no. 895412 to D.K.); a Helen Hay Whitney Foundation Postdoctoral Fellowship (to M.E.W.); the Howard Hughes Medical Institute (to M.E.W.) and a Research Fellowship at Gonville and Caius College of Cambridge University (to V.V.). The funders had no role in study design, data collection and analysis, decision to publish or preparation of the manuscript. For the purpose of open access, the MRC Laboratory of Molecular Biology has applied a CC BY public copyright licence to any Author Accepted Manuscript version arising.

## Author contributions

D.K. designed and implemented Blush regularization, ran most experiments and analyzed the results. K.J. contributed to data preprocessing. M.E.W. contributed and analysed the Aca2–RNA dataset, and contributed to analysis of the PfCRT dataset. S.L. contributed the FIA dataset. V.V. contributed the Ste2 dataset. T.N. analysed the results. S.H.W.S. supervised the project and contributed to RELION integration. All authors contributed to the writing of the manuscript.

## Competing interests

The authors declare no competing interests.

## Additional information

**Extended data** is available for this paper at https://doi.org/10.1038/s41592-024-02304-8.

**Correspondence and requests for materials** should be addressed to Dari Kimanius or Sjors H. W. Scheres.

**Extended Data Table 1 | Cryo-EM datasets of Ste2 and Aca2. Details on cryo-EM data collection and processing of the Ste2 and Aca2 datasets are presented**

| | Ste2 | Aca2 |
|---|---|---|
| **Data collection** | | |
| Microscope | Thermo Scientific Krios G3i | Thermo Scientific Krios G3i |
| Camera | Falcon 4i | Gatan K3 |
| Magnification | 96,000x | 165,000x |
| Voltage (kV) | 300 | 300 |
| Pixel size (Å) | 0.824 | 0.531 |
| Total electron exposure (electron/Å$^2$) | 56 | 72 |
| Exposure rate (electron/pixel/sec) | 7.9 | 15 |
| Number of hardware frames during exposure | 1539 | N/A |
| Defocus range ($\mu$m) | 0.8 - 2.4 | 0.8 - 1.5 |
| Automation software | EPU | EPU |
| Energy filter slit width | N/A | 20 eV |
| Number of micrographs collected | 13,254 | 11,232 |
| | | |
| **3D reconstruction** | | |
| Number of frames for motion correction | 70 | 40 |
| Number of extracted particles | 2,608,181 | 6,449,613 |
| Number of particles in final map | 147,797 | 301,833 |
| Symmetry imposed | C2 | C1 |
| Resolution (Å; at FSC=0.143; unmasked) | 3.9 | 2.8 |
| Resolution (Å; at FSC=0.143; masked) | 3.4 | 2.6 |
| Sharpening B-factor (Å$^2$) | -160 | -90 |

# Reporting Summary

## Statistics

For all statistical analyses, confirm that the following items are present in the figure legend, table legend, main text, or Methods section.

| n/a | Confirmed | |
|---|---|---|
| ☐ | ☒ | The exact sample size (*n*) for each experimental group/condition, given as a discrete number and unit of measurement |
| ☒ | ☐ | A statement on whether measurements were taken from distinct samples or whether the same sample was measured repeatedly |
| ☒ | ☐ | The statistical test(s) used AND whether they are one- or two-sided<br>*Only common tests should be described solely by name; describe more complex techniques in the Methods section.* |
| ☒ | ☐ | A description of all covariates tested |
| ☒ | ☐ | A description of any assumptions or corrections, such as tests of normality and adjustment for multiple comparisons |
| ☒ | ☐ | A full description of the statistical parameters including central tendency (e.g. means) or other basic estimates (e.g. regression coefficient) AND variation (e.g. standard deviation) or associated estimates of uncertainty (e.g. confidence intervals) |
| ☒ | ☐ | For null hypothesis testing, the test statistic (e.g. *F*, *t*, *r*) with confidence intervals, effect sizes, degrees of freedom and *P* value noted<br>*Give P values as exact values whenever suitable.* |
| ☒ | ☐ | For Bayesian analysis, information on the choice of priors and Markov chain Monte Carlo settings |
| ☒ | ☐ | For hierarchical and complex designs, identification of the appropriate level for tests and full reporting of outcomes |
| ☒ | ☐ | Estimates of effect sizes (e.g. Cohen's *d*, Pearson's *r*), indicating how they were calculated |

*Our web collection on statistics for biologists contains articles on many of the points above.*

## Software and code

Policy information about availability of computer code

| Data collection | We used publicly available data from EMPIAR, plus a new data set on Aca2:RNA that will be published in an independent paper elsewhere. |
|---|---|
| Data analysis | RELION is open-source (GPLv2 license) and can be downloaded from https://github.com/3dem. We used RELION version 5.0-beta-1. Parts of the code of Blush were written in pytorch (we used version 2.0.1). Code used in the training of the Blush denoiser is available from https://github.com/dkimanius/blush-training. |

For manuscripts utilizing custom algorithms or software that are central to the research but not yet described in published literature, software must be made available to editors and reviewers. We strongly encourage code deposition in a community repository (e.g. GitHub). See the Nature Portfolio guidelines for submitting code & software for further information.

## Data

Policy information about availability of data

All manuscripts must include a data availability statement. This statement should provide the following information, where applicable:
- Accession codes, unique identifiers, or web links for publicly available datasets
- A description of any restrictions on data availability
- For clinical datasets or third party data, please ensure that the statement adheres to our policy

A list of EMDB entries used for training, plus manually curated masks can be downloaded from https://zenodo.org/records/10553452. We used publicly available

cryo-EM data sets from the EMPIAR data base (accession numbers 10330, 10180 and 11720). The Aca2:RNA data set as been uploaded to EMPIAR (accession number 11918).

## Human research participants

Policy information about studies involving human research participants and Sex and Gender in Research.

| Reporting on sex and gender | N/A |
| --- | --- |
| Population characteristics | N/A |
| Recruitment | N/A |
| Ethics oversight | N/A |

Note that full information on the approval of the study protocol must also be provided in the manuscript.

# Field-specific reporting

Please select the one below that is the best fit for your research. If you are not sure, read the appropriate sections before making your selection.

☒ Life sciences       ☐ Behavioural & social sciences       ☐ Ecological, evolutionary & environmental sciences

For a reference copy of the document with all sections, see nature.com/documents/nr-reporting-summary-flat.pdf

# Life sciences study design

All studies must disclose on these points even when the disclosure is negative.

| Sample size | Cryo-EM data set sizes were determined by the publicly available data sets from EMPIAR, or by the number of acquired micrographs for the Ste2 and Aca2:RNA data sets. |
| --- | --- |
| Data exclusions | We used previously published data sets in their entirety. For the Aca2:RNA data set, we performed 3D classification as indicated in figure 6 to select the particles giving rise to the final map. |
| Replication | No replication experiments were performed, as noise on the outcome of the computational analysis was not considered to be affecting the conclusions. |
| Randomization | For each data set, two random half-sets were employed for resolution estimation. Randomisation was performed using a random number generator. |
| Blinding | Blinding is not performed in cryo-EM analysis, because knowledge of the target by the experimentalist is not expected to affect the outcome of the reconstruction algorithm. |

# Reporting for specific materials, systems and methods

We require information from authors about some types of materials, experimental systems and methods used in many studies. Here, indicate whether each material, system or method listed is relevant to your study. If you are not sure if a list item applies to your research, read the appropriate section before selecting a response.

## Materials & experimental systems

| n/a | Involved in the study |
| --- | --- |
| ☒ ☐ | Antibodies |
| ☒ ☐ | Eukaryotic cell lines |
| ☒ ☐ | Palaeontology and archaeology |
| ☒ ☐ | Animals and other organisms |
| ☒ ☐ | Clinical data |
| ☒ ☐ | Dual use research of concern |

## Methods

| n/a | Involved in the study |
| --- | --- |
| ☒ ☐ | ChIP-seq |
| ☒ ☐ | Flow cytometry |
| ☒ ☐ | MRI-based neuroimaging |

