## [Peer Review File · Nature Methods]

Peer Review Information

Manuscript Title: Data-driven regularisation lowers the size barrier of cryo-EM structure determination

Corresponding author name(s): Sjors Scheres, Dari Kimanius

Editorial Notes: None

Reviewer Comments & Decisions:

Decision Letter, initial version:

Dear Sjors,

Your Article, "Data-driven regularisation lowers the size barrier of cryo-EM structure determination", has now been seen by 2 reviewers. As you will see from their comments below, although the reviewers find your work of considerable potential interest, they have raised a number of concerns. We are interested in the possibility of publishing your paper in Nature Methods, but would like to consider your response to these concerns before we reach a final decision on publication.

We therefore invite you to revise your manuscript to address these concerns. In particular, we would like you to address the concerns about the novelty of the approach, general applicability to both homogeneous and heterogeneous reconstruction, and comparisons/benchmarking with M.

- * include a point-by-point response to the reviewers and to any editorial suggestions
- * please underline/highlight any additions to the text or areas with other significant changes to facilitate review of the revised manuscript
- * address the points listed described below to conform to our open science requirements
- * ensure it complies with our general format requirements as set out in our guide to authors at www.nature.com/naturemethods

* resubmit all the necessary files electronically by using the link below to access your home page

[Redacted]

We hope to receive your revised paper within 6 weeks. If you cannot send it within this time, please let us know. In this event, we will still be happy to reconsider your paper at a later date so long as nothing similar has been accepted for publication at Nature Methods or published elsewhere.

OPEN SCIENCE REQUIREMENTS

REPORTING SUMMARY AND EDITORIAL POLICY CHECKLISTS

DATA AVAILABILITY

We strongly encourage you to deposit all new data associated with the paper in a persistent repository where they can be freely and enduringly accessed. We recommend submitting the data to discipline-specific and community-recognized repositories; a list of repositories is provided here:

<http://www.nature.com/sdata/policies/repositories>

All novel DNA and RNA sequencing data, protein sequences, genetic polymorphisms, linked genotype

and phenotype data, gene expression data, macromolecular structures, and proteomics data must be deposited in a publicly accessible database, and accession codes and associated hyperlinks must be provided in the "Data Availability" section.

CODE AVAILABILITY

Please include a "Code Availability" subsection in the Online Methods which details how your custom code is made available. Only in rare cases (where code is not central to the main conclusions of the paper) is the statement "available upon request" allowed (and reasons should be specified).

For more information on our code sharing policy and requirements, please see: <https://www.nature.com/nature-research/editorial-policies/reporting-standards#availability-of-computer-code>

MATERIALS AVAILABILITY

Authors reporting new chemical compounds must provide chemical structure, synthesis and

characterization details. Authors reporting mutant strains and cell lines are strongly encouraged to use established public repositories.

SUPPLEMENTARY PROTOCOL

To help facilitate reproducibility and uptake of your method, we ask you to prepare a step-by-step Supplementary Protocol for the method described in this paper. We encourage authors to share their step-by-step experimental protocols on a protocol sharing platform of their choice and report the protocol DOI in the reference list. Nature Portfolio 's Protocol Exchange is a free-to-use and open resource for protocols; protocols deposited in Protocol Exchange are citable and can be linked from the published article. More details can found at www.nature.com/protocolexchange/about.

ORCID

Nature Methods is committed to improving transparency in authorship. As part of our efforts in this direction, we are now requesting that all authors identified as 'corresponding author' on published papers create and link their Open Researcher and Contributor Identifier (ORCID) with their account on the Manuscript Tracking System (MTS), prior to acceptance. This applies to primary research papers only. ORCID helps the scientific community achieve unambiguous attribution of all scholarly contributions. You can create and link your ORCID from the home page of the MTS by clicking on 'Modify my Springer Nature account'. For more information please visit please visit www.springernature.com/orcid.

Sincerely,
Arunima

Arunima Singh, Ph.D.
Senior Editor
Nature Methods

Reviewers' Comments:

Reviewer #1:
None

Reviewer #2:
Remarks to the Author:

This paper introduced Blush regularisation for single-particle cryo-EM structure refinement, which has the ability to improve the structure resolution, particularly for small proteins. The regularisation is achieved by training a denoising convolutional neural network on pairs of half-set maps obtained from the EMDB server and repurposing the trained neural network to provide prior knowledge in the iterative structure refinement process. Experimental results suggest that such unique regularisation helps achieve better resolution with single-particle reconstruction with RELION, particularly in low SNR settings. Such regularisation successfully recovered the structure of the Aca2 protein (of size 40kDa) and amyloid with only 30 ordered core residues, which was not possible with the previous RELION refinement procedure.

Despite demonstrating multiple successful contributions and huge promise, I have a few major and minor concerns regarding the paper, which are mentioned below:

Major concerns:

1. Previous works [1] have demonstrated the reconstruction of membrane proteins of 23 kDa size from single-particle cryo-EM images. In such a case, recovering the structure of Aca2 protein of 40kDa size though remarkable but not surprising. The authors should provide a detailed discussion regarding the biological contribution it brought that was not possible before.
2. Previous works [2] have already used trained denoising networks to refine cryo-EM reconstructions. As a result, the novelty and contribution of the proposed approach of Blush regularisation is not very clear. The authors did mention that, [2] did not express general prior knowledge of cryo-EM refinements unlike Blush, but the statement is not backed by experimental results.
3. The effectiveness of the proposed regularisation depends on choosing a hyperparameter ρ , which is defined as the resolution where solvent-corrected FSC drops below 0.143. How generalizable is such a selection? Will such selection generalize for single-particle cryo-EM reconstructions other than those mentioned in the paper?

[1] Wu, X., & Rapoport, T. A. (2021). Cryo-EM structure determination of small proteins by nanobody-binding scaffolds (Legobodies). *Proceedings of the National Academy of Sciences*, 118(41), e2115001118.

[2] Tegunov, D., Xue, L., Dienemann, C., Cramer, P., & Mahamid, J. (2021). Multi-particle cryo-EM refinement with M visualizes ribosome-antibiotic complex at 3.5 Å in cells. *Nature Methods*, 18(2), 186-193.

Minor concerns:

1. The proposed regularisation is only used for the homogeneous reconstruction of single-particle cryo-EM images with RELION. Will it be as useful for heterogeneous reconstruction with cryoSPARC, 3DFlex, cryoDRGN, e2gmm, etc. methods?
2. To train the denoising convolution neural network, the authors selected 422 pairs of half-maps from the EMDB server. The authors mentioned that they only kept the half-maps over 4 Å resolutions. Are there only 422 pairs of half-maps with such a resolution in the EMDB server? I think there might be more. In such a case, how were these 422 pairs selected? Also, how do these selection choices affect

the regularisation performance of the trained denoising network? The authors may provide more discussion into this for a better understanding of the method design.

Reviewer #3:

Remarks to the Author:

The paper presents a data-driven approach to "regularization" expectation maximization. The algorithm has been implemented in RELION 5. The experiments are informative, and the results are compelling. The authors demonstrate that the algorithm can be useful in the analysis of relatively small molecules and in increasing the resolution in analysis more broadly. They also demonstrate that the algorithm can be used to denoise reconstructions.

The authors take care in integrating these deep learning algorithms in a way that potentially reduces the risks of overfitting and hallucinations.

The paper is well-written, and it conveys the ideas and significance well.

The algorithm is interesting in its own right. It is especially important as a component in the next version of the heavily used RELION software.

The paper is very good as is. My suggestions and questions are relatively minor.

Training:

Is the code posted with RELION5 sufficient for reproducing the training? Are the parameters used in the random augmentation provided there? Is the list of EMDB half-maps available there? I do not think these are crucial, but can be good.

Were these 422 pairs of maps, each augmented to (how many) training pairs?

Page 3, difficulty in the solvent area. Can the authors elaborate more about the issue encountered in the solvent area? Is it the case that the authors tried a simpler regularizer without the special treatment of the solvent and found that it did not work well?

Denoiser architecture:

Do I understand correctly that the denoiser operates in the original domain, not the Fourier domain?

Is the denoiser restricted to 64^3 volumes? Does this create potential difficulties?

How are volumes of size and voxel size different than here handled in RELION 5.

Formal description:

The description on page 4 refers to the algorithm as a replacement for a Wiener filter, and it is also referred to as a "regularizer".

The authors may correct me if I am wrong, but the Bayesian Expectation maximization approach uses a prior that, in the RELION implementation, bears a resemblance to the Wiener filter. But, it is not exactly a filter that can be exchanged with any generalized filter, at least not without tailoring the meaning and structure of the prior. For example, a prior would usually have a smaller effect when there is more data, whereas a filter that depends only on the reconstructed volume would not necessarily have this property.

Is the original RELION prior still used in the reconstruction of the half-volumes before the denoiser is used?

Does the addition of a filter not deviate formally from the expectation-maximization approach? Is there a way to formally express this filter as a prior?

In case this deviates from expectation maximization, I think it is appropriate to make a remark somewhere. It does not mean that the algorithm is not very useful.

The description of "spectral trailing" in the "Iterative denoising..." section:

"Fourier shells". The term can be ambiguous. What do the authors mean? It is just a technical detail but it is useful in understanding the filter.

"...Nyquist frequency of the denoiser, here set to 3A...". Is this a setting or an implication of the rescaling to 1.5A?

The text in this section suggests that ρ is determined in each iteration by the FSC cutoff. On page 5, it seems to be set to a certain value. Do I understand correctly that in the second case, it was set manually for the demonstration of filtering a reconstruction?

The description in this section is OK, but the authors may wish to make it a little more clear.

Page 6: "ase" -> "case"?

High-frequency information: Page 7: "Spectral trailing ensured that no information from the denoiser was inserted beyond 3.7A resolution." and page 10: "Because Fourier components... cannot result in hallucination..."

I wonder if this is a completely accurate statement as "information" could come in through the alignment (perhaps reminiscent of the Einstein from noise problem). I do not think that these nuances need to be treated here, so I do not think the statement on page 7 is problematic in the context where it appears. The statement on page 10 can be toned down a little.

Discussion: "as a result, the FSC..." This sentence is not very clear but seems to be important. Can the authors elaborate?

It could be worth noting what language or deep learning framework was used in the implementation.

Author Rebuttal to Initial comments
--

We thank both reviewers for their constructive remarks. Our replies are in blue below.

Reviewer #2:

This paper introduced Blush regularisation for single-particle cryo-EM structure refinement, which has the ability to improve the structure resolution, particularly for small proteins. The regularisation is achieved by training a denoising convolutional neural network on pairs of half-set maps obtained from the EMDB server and repurposing the trained neural network to provide prior knowledge in the iterative structure refinement process. Experimental results suggest that such unique regularisation helps achieve better resolution with single-particle reconstruction with RELION, particularly in low SNR settings. Such regularisation successfully recovered the structure of the Aca2 protein (of size 40kDa) and amyloid with only 30 ordered core residues, which was not possible with the previous RELION refinement procedure.

Despite demonstrating multiple successful contributions and huge promise, I have a few major and minor concerns regarding the paper, which are mentioned below:

Major concerns:

1. Previous works [1] have demonstrated the reconstruction of membrane proteins of 23 kDa size from single-particle cryo-EM images. In such a case, recovering the structure of Aca2 protein of 40kDa size though remarkable but not surprising. The authors should provide a detailed discussion regarding the biological contribution it brought that was not possible before.

The 23 kDa size mentioned above was not the molecular weight of the complex that was solved by cryo-EM in that study. Instead, in order to bring the size to one suitable for cryo-EM structure determination, the 23 kDa protein was attached to so-called Lego-bodies, Fab-fragments combined with other proteins. In the introduction section, we have included a citation to this work and explained the difference with our approach. The new sentences read:

"The ill-posedness of the reconstruction imposes a minimum size barrier for cryo-EM structure determination, because smaller complexes yield images with lower signal-to-noise ratios. Although this barrier may be overcome experimentally by complex formation of small targets with other proteins [Wu,Rapoport], the formation of sufficiently rigid complexes is often difficult. Here, we explore a computational method that lowers the size barrier for existing cryo-EM data sets"

2. Previous works [2] have already used trained denoising networks to refine cryo-EM reconstructions. As a result, the novelty and contribution of the proposed approach of Blush regularisation is not very clear. The authors did mention that, [2] did not express general prior knowledge of cryo-EM refinements unlike Blush, but the statement is not backed by experimental results.

The M approach and Blush are fundamentally different in that the former trains a new neural network on each cryo-EM reconstruction to which it is applied, and as such cannot learn general features about cryo-EM reconstructions beyond those in the reconstruction at hand. The observation that our single, pre-trained network improves a wide variety of use cases at least suggests that it has indeed learnt useful features about cryo-EM reconstructions in general. Other differences between the approach in M and our approach lie in the way we train our denoiser, by complementing the unsupervised noise2noise approach with various supervised elements. An interesting future direction might be to investigate a hybrid of the two methods, where our pre-trained model is fine-tuned for each specific data set using the method in M.

We have rephrased the corresponding section in the Discussion to include these points. It now reads:

“A previous approach that used noise2noise, implemented in the M software [27], trains a new neural network for each data set to which it is applied, using only half-maps from that same data set. As such, the neural network in the M software can only learn features that are specific to the data set at hand. In contrast, we pre-train a single neural network on a diverse set of high-resolution half-maps from the EMDB. The observation that our pre-trained network improves cryo-EM reconstructions for a wide variety of macromolecular complexes suggests that it has learnt useful features about cryo-EM structures in general. In addition, although our approach was inspired by noise2noise, it blends the unsupervised elements from noise2noise training with novel application-specific elements, such as recycling and supervised masks in Fourier-space and in real-space. An interesting avenue for future research may be to explore a combination of the two approaches, where the pre-trained Blush network is fine-tuned using the half-maps of the reconstruction at hand, using techniques similar to those implemented in M.”

3. The effectiveness of the proposed regularisation depends on choosing a hyperparameter ρ , which is defined as the resolution where solvent-corrected FSC drops below 0.143. How generalizable is such a selection? Will such selection generalize for single-particle cryo-EM reconstructions other than those mentioned in the paper?

Although the FSC=0.143 criterion is well established for resolution assessment of cryo-EM structures, the reviewer correctly identifies the heuristic of spectral trailing (and the choice of its parameter ρ) as a limitation of our method. We illustrate this ourselves for the PfCRT case, where the model-to-map FSC actually improves when one does not use spectral trailing (Fig 2c). Although better methods may exist to guarantee the reliability of the resolution estimates, our approach of spectral trailing represents the first effort to explicitly recognize the dangers of over-estimating resolution when using information-rich priors in cryo-EM reconstruction. As such, we consider spectral trailing to be an important part of the paper. To clarify these points, we have added the following statement to the Discussion section:

“Although spectral trailing represents the first attempt to prevent over-estimation of resolution when using information-rich priors in cryo-EM reconstruction, it may not be the optimal

solution. In fact, as exemplified by the PfCRT dataset (Fig 2), spectral trailing can lead to under-estimation of resolution. Future exploration of the damping effect of the network in Fourier space may lead to better approaches to safeguard against over-estimation of resolution."

[1] Wu, X., & Rapoport, T. A. (2021). Cryo-EM structure determination of small proteins by nanobody-binding scaffolds (Legobodies). *Proceedings of the National Academy of Sciences*, 118(41), e2115001118.

[2] Tegunov, D., Xue, L., Dienemann, C., Cramer, P., & Mahamid, J. (2021). Multi-particle cryo-EM refinement with M visualizes ribosome-antibiotic complex at 3.5 Å in cells. *Nature Methods*, 18(2), 186-193.

Minor concerns:

1. The proposed regularisation is only used for the homogeneous reconstruction of single-particle cryo-EM images with RELION. Will it be as useful for heterogeneous reconstruction with cryoSPARC, 3DFlex, cryoDRGN, e2gmm, etc. methods?

We respectfully disagree. Our approach was demonstrated with two different methods that handle heterogeneous data: multi-body refinement of the spliceosome (Fig 4) and 3D classification for Aca2 (Fig 6). In both cases, Blush improved the handling of structural heterogeneity. Furthermore, Ste2 exhibits extensive heterogeneity in large portions of the structure, and the density inside the detergent regions of Ste2 and PfCRT are also highly heterogeneous. To better emphasise the observation that 3D classification of the Aca2 data set did not work without Blush regularisation, we added the following sentence to the Results section:

"Similar 3D classifications without Blush regularisation did not yield recognizable protein features."

Any method that allows real-space priors could in principle benefit from our approach. Out of the methods that the reviewer has listed, 3DFlex and 3DVA in cryoSPARC are among such methods (cryoDRGN works explicitly in Fourier space and e2gmm uses 3D Gaussian pseudo-atoms). To make this broad applicability of our method clearer, we added the following sentence to the Discussion:

"In fact, provided that they allow real-space constraints, Blush regularisation may improve a wide range of cryo-EM methods, ranging from standard refinement approaches in alternative software packages to approaches for dealing specifically with structural heterogeneity, e.g. [30–32]."

2. To train the denoising convolutional neural network, the authors selected 422 pairs of half-maps from the EMDB server. The authors mentioned that they only kept the half-maps over 4 Å resolutions. Are there only 422 pairs of half-maps with such a resolution in the EMDB server? I think there might be more. In such a case, how were these 422 pairs selected? Also, how do these selection choices affect the regularisation performance of the

trained denoising network? The authors may provide more discussion into this for a better understanding of the method design.

Not so many EMDB entries contain half-maps, and for a long time these have not been annotated explicitly. The 4 Å resolution cutoff, combined with manual curation to remove maps with artefacts and maps with the same structure as other entries in the training set, led to the final number of 422. We now explicitly mention this in the Methods section:

"Maps with obvious artefacts, for example those associated with overfitting, and maps of the same structure as already present in the training set, were eliminated by manual curation."

Reviewer #3:

The paper presents a data-driven approach to "regularization" expectation maximization. The algorithm has been implemented in RELION 5. The experiments are informative, and the results are compelling. The authors demonstrate that the algorithm can be useful in the analysis of relatively small molecules and in increasing the resolution in analysis more broadly. They also demonstrate that the algorithm can be used to denoise reconstructions. The authors take care in integrating these deep learning algorithms in a way that potentially reduces the risks of overfitting and hallucinations. The paper is well-written, and it conveys the ideas and significance well.

The algorithm is interesting in its own right. It is especially important as a component in the next version of the heavily used RELION software. The paper is very good as is. My suggestions and questions are relatively minor.

We are grateful for these comments. :-)

Training:

Is the code posted with RELION5 sufficient for reproducing the training? Are the parameters used in the random augmentation provided there? Is the list of EMDB half-maps available there? I do not think these are crucial, but can be good. Were these 422 pairs of maps, each augmented to (how many) training pairs?

We have now added a new section, Data Availability, that contains the following text:

"The full list of EMDB entries that were used to train the denoiser, along with the manually curated masks, along with the manually curated masks can be downloaded from <https://zenodo.org/records/10553452>."

Furthermore, under Code availability we have added:

"Additionally, code used in the training procedure of the Blush denoiser model is available on <https://github.com/dkimanius/blush-training>."

Page 3, difficulty in the solvent area. Can the authors elaborate more about the issue encountered in the solvent area? Is it the case that the authors tried a simpler regularizer without the special treatment of the solvent and found that it did not work well?

Yes. Initially, we trained a model without real-space masking and found that it could, to some extent, learn to smoothen disordered regions, including detergent micelles and solvent regions, leading to improvements in alignment. Therefore, we decided to further amplify this effect using supervised masking methods, resulting in further improvements. We have added the following statement to the Methods section:

“Although initial versions of our training protocol did not include masks, we observed that the resulting networks would learn to smoothen densities in disordered regions, like the solvent or detergent micelles, which would improve image alignments. To amplify these effects, we then implemented the supervised masking approach with M_i and $h(\cdot)$.”

Denoiser architecture: Do I understand correctly that the denoiser operates in the original domain, not the Fourier domain? Is the denoiser restricted to 64^3 volumes? Does this create potential difficulties? How are volumes of size and voxel size different than here handled in RELION 5.

Yes, that is correct. The denoiser is trained on many 64^3 windows that are cropped from larger maps. When applying the denoiser on such larger maps, a window is moved over the entire map. Additionally, during reconstruction, we rescale the voxel size to 1.5 Å, apply the denoiser, and rescale it back to the original voxel size. We have made this more explicit in the Methods section by adding the following statement:

“By re-scaling all maps to a common voxel size of 1.5 Å, and then cropping maps to patches of 64^3 voxels, the network can be trained on and applied to maps of any size. For application of the denoiser to maps that are larger than one patch, overlapping patches are denoised independently.”

Formal description: The description on page 4 refers to the algorithm as a replacement for a Wiener filter, and it is also referred to as a “regularizer”. The authors may correct me if I am wrong, but the Bayesian Expectation maximization approach uses a prior that, in the RELION implementation, bears a resemblance to the Wiener filter. But, it is not exactly a filter that can be exchanged with any generalized filter, at least not without tailoring the meaning and structure of the prior. For example, a prior would usually have a smaller effect when there is more data, whereas a filter that depends only on the reconstructed volume would not necessarily have this property. Is the original RELION prior still used in the reconstruction of the half-volumes before the denoiser is used? Does the addition of a filter not deviate formally from the expectation-maximization approach? Is there a way to formally express this filter as a prior? In case this deviates from expectation maximization, I think it is appropriate to make a remark somewhere. It does not mean that the algorithm is not very useful.

In our proof-of-concept paper (Kimanius et al, IUCrJ, 2021), we presented a more detailed argument on how the denoiser network functions as a regularizer, specifically in section 2.3 of that publication. In the current work, we do not claim that the denoiser acts as an explicit prior within a strict Bayesian framework. Instead based on empirical observations, we notice that the denoiser acts as a regulariser in a generalised sense, effectively replacing the filter

operation in the M-step. Compared to the approach in our proof-of-concept paper, we also noticed that the explicit tuning of the regularisation parameter τ was superfluous in Blush regularisation (which approximates to setting τ to zero in equation 17 in the proof-of-concept paper).

We have added the following statement to the Methods section to make it clearer that we do not make explicit claims about the denoiser in a Bayesian framework:

"We refer to the application of our pre-trained denoiser within the iterative expectation-maximisation algorithm as Blush regularisation. In our original work, with simulated data, we incorporated the denoiser into the L_2 regularisation in the M-step, based on the approximation that the prior function is "close" to a Gaussian [6]. In this work, we do not make formal claims about the role of the denoiser within a Bayesian framework. Rather, our approach is motivated by empirical observations."

The description of "spectral trailing" in the "iterative denoising..." section: "Fourier shells". The term can be ambiguous. What do the authors mean? It is just a technical detail but it is useful in understanding the filter. "

These are the same as the shells in the Fourier Shell Correlation. To avoid confusion we have added a subclause: "(each shell is one Fourier voxel wide)".

...Nyquist frequency of the denoiser, here set to $3A$...". Is this a setting or an implication of the rescaling to $1.5A$?

This is an implication of the rescaling. As outlined above, re-scaling is now mentioned more explicitly in the Methods section.

The text in this section suggests that ρ is determined in each iteration by the FSC cutoff. On page 5, it seems to be set to a certain value. Do I understand correctly that in the second case, it was set manually for the demonstration of filtering a reconstruction? The description in this section is OK, but the authors may wish to make it a little more clear.

Our original statement on page 5 was unclear. The threshold indeed changes at every iteration, and the statement refers to the last iteration. To avoid confusion, we have modified the statement to:

"In the last iteration, spectral trailing was applied with a cut-off at 3.5 \AA , beyond which no information from the denoiser was used."

Page 6: "ase" -> "case"?
Corrected.

High-frequency information: Page 7: "Spectral trailing ensured that no information from the denoiser was inserted beyond 3.7 \AA resolution." and page 10: "Because Fourier components... cannot result in hallucination..."

I wonder if this is a completely accurate statement as "information" could come in through the alignment (perhaps reminiscent of the Einstein from noise problem). I do not think that these nuances need to be treated here, so I do not think the statement on page 7 is problematic in the context where it appears. The statement on page 10 can be toned down a little.

We have limited the scope of the statement on page 10. It now reads:

"Because Fourier components near the resolution estimate of the final map will not have been affected by the denoiser, over-estimation of resolution due to the denoiser is not possible."

Discussion: "as a result, the FSC..." This sentence is not very clear but seems to be important. Can the authors elaborate?

This statement reflects the change in the algorithm that the 3D Wiener filter of the approach with the L2-regulariser has now been replaced by the denoising network. We have rephrased the sentence to:

"Within the framework of Blush regularisation, the denoiser replaces the filter operation that constrains the power of Fourier-space components in the baseline algorithm. As a result, the FSC between independently refined subsets is no longer used to define a 3D Wiener filter that is applied to the intermediate reconstructions."

It could be worth noting what language or deep learning framework was used in the implementation.

We now mention the framework (pyTorch) in the Methods section.

Decision Letter, first revision:

Dear Sjors,

Thank you for submitting your revised manuscript "Data-driven regularisation lowers the size barrier of cryo-EM structure determination" (N METH-A54283A). It has now been seen by the original referees and their comments are below. The reviewers find that the paper has improved in revision, and therefore we'll be happy in principle to publish it in Nature Methods, pending minor revisions to satisfy the referees' final requests and to comply with our editorial and formatting guidelines.

TRANSPARENT PEER REVIEW

Please note: we allow redactions to authors' rebuttal and reviewer comments in the interest of confidentiality. If you are concerned about the release of confidential data, please let us know specifically what information you would like to have removed. Please note that we cannot incorporate redactions for any other reasons. Reviewer names will be published in the peer review files if the reviewer signed the comments to authors, or if reviewers explicitly agree to release their name. For more information, please refer to our FAQ page.

ORCID

Sincerely,
Arunima

Arunima Singh, Ph.D.
Senior Editor
Nature Methods

Reviewer #2 (Remarks to the Author):

The authors adequately addressed all of my concerns and confusions. They have clarified the most impactful contribution of the paper, reducing the size barrier of cryo-EM structure determination through a computational approach, which was not done before. I think the manuscript and the software integrated into RELION will significantly facilitate structure recovery from single-particle cryo-EM datasets, particularly for very small particles. Therefore, I recommend accepting the manuscript.

Reviewer #3 (Remarks to the Author):

The authors addressed most of my concerns.

I have one remaining formal concern, which is unlikely to have significant practical implications. The authors argue the FSC over-estimation due to the denoiser is "not possible." "Not possible" is a very strong assertion.

Suppose that the denoiser somehow turns the volume to something roughly spherically symmetric. It is conceivable that subsequent iterations reconstruct roughly spherically symmetric volumes, and those would have relatively high FSC. Therefore, it is technically possible to get FSC overestimation due to a denoiser (although not in the way the authors meant, and it may depend on what the FSC is supposed to measure here). So, while I follow the author's logic, I think that "not possible" might not be formally accurate. I think it would be best to rephrase that sentence a little more carefully. Perhaps saying something like "unlikely" or even limiting the scope by saying that it cannot happen "directly" would be more accurate and not too distracting for the reader.

As before, I think that the paper can be published as is. This small correction would improve it, but it has little practical significance.

Author Rebuttal, first revision:

Reviewer #2:

Remarks to the Author:

The authors adequately addressed all of my concerns and confusions. They have clarified the most impactful contribution of the paper, reducing the size barrier of cryo-EM structure determination through a computational approach, which was not done before. I think the manuscript and the software integrated into RELION will significantly facilitate structure recovery from single-particle cryo-EM datasets, particularly for very small particles. Therefore, I recommend accepting the manuscript.

Reviewer #3:

Remarks to the Author:

The authors addressed most of my concerns.

I have one remaining formal concern, which is unlikely to have significant practical implications. The authors argue the FSC over-estimation due to the denoiser is "not possible." "Not possible" is a very strong assertion.

Suppose that the denoiser somehow turns the volume to something roughly spherically symmetric. It is conceivable that subsequent iterations reconstruct roughly spherically symmetric volumes, and those would have relatively high FSC. Therefore, it is technically possible to get FSC overestimation due to a denoiser (although not in the way the authors meant, and it may depend on what the FSC is supposed to measure here). So, while I follow the author's logic, I think that "not possible" might not be formally accurate. I think it would be best to rephrase that sentence a little more carefully. Perhaps saying something like "unlikely" or even limiting the scope by saying that it cannot happen "directly" would be more accurate and not too distracting for the reader.

Following the reviewer's suggestion, we have replaced "is not possible" by "cannot happen directly".

Final Decision Letter:

Dear Sjors,

I am pleased to inform you that your Article, "Data-driven regularisation lowers the size barrier of cryo-EM structure determination", has now been accepted for publication in Nature Methods. The received and accepted dates will be October 27, 2023 and May 8, 2024. This note is intended to let you know what to expect from us over the next month or so, and to let you know where to address any further questions.

Over the next few weeks, your paper will be copyedited to ensure that it conforms to Nature Methods style. Once your paper is typeset, you will receive an email with a link to choose the appropriate publishing options for your paper and our Author Services team will be in touch regarding any additional information that may be required. It is extremely important that you let us know now whether you will be difficult to contact over the next month. If this is the case, we ask that you send us the contact information (email, phone and fax) of someone who will be able to check the proofs and deal with any last-minute problems.

After the grant of rights is completed, you will receive a link to your electronic proof via email with a request to make any corrections within 48 hours. If, when you receive your proof, you cannot meet

this deadline, please inform us at rjsproduction@springernature.com immediately.

Please note that *Nature Methods* is a Transformative Journal (TJ). Authors may publish their research with us through the traditional subscription access route or make their paper immediately open access through payment of an article-processing charge (APC). Authors will not be required to make a final decision about access to their article until it has been accepted. Find out more about Transformative Journals

If you are active on Twitter/X, please e-mail me your and your coauthors' handles so that we may tag you when the paper is published.

Please note that you and any of your coauthors will be able to order reprints and single copies of the issue containing your article through Nature Portfolio's reprint website, which is located at <http://www.nature.com/reprints/author-reprints.html>. If there are any questions about reprints please

send an email to author-reprints@nature.com and someone will assist you.

Best regards,
Arunima

Arunima Singh, Ph.D.
Senior Editor
Nature Methods